# Unraveling the Complex Interplay Between Neuroinflammation and Depression: A Comprehensive Review

**DOI:** 10.3390/ijms26041645

**Published:** 2025-02-14

**Authors:** Andreea Sălcudean, Ramona-Amina Popovici, Dana Emanuela Pitic, Diana Sârbu, Adela Boroghina, Mohammad Jomaa, Matin Asad Salehi, Alsayed Ahmad Mhd Kher, Maria Melania Lica, Cristina Raluca Bodo, Virgil Radu Enatescu

**Affiliations:** 1Department of Ethics and Social Sciences, George Emil Palade University of Medicine, Pharmacy, Science and Technology of Targu Mures, 540142 Targu Mures, Romania; andreea.salcudean@umfst.ro (A.S.); melaniacozma76@gmail.com (M.M.L.); cristina.bodo@umfst.ro (C.R.B.); 2Department of Management and Communication in Dental Medicine, Department I, Faculty of Dental Medicine, Victor Babes University of Medicine and Pharmacy of Timisoara, 9 Revolutiei 1989 Bv., 300070 Timisoara, Romania; dana.pitic@umft.ro; 3Doctoral School of Pharmacy, Victor Babes University of Medicine and Pharmacy of Timisoara, 2 Eftimie Murgu Square, 300041 Timisoara, Romania; sarbu.diana@umft.ro; 4Doctoral School of Dental Medicine, Victor Babes University of Medicine and Pharmacy of Timisoara, 9 Revolutiei 1989 Bv., 300070 Timisoara, Romania; adela.boroghina@umft.ro (A.B.); mohammad.jomaa@umft.ro (M.J.); matin.asad-salehi@umft.ro (M.A.S.); mhd.alsaeyd-ahmad@umft.ro (A.A.M.K.); 5Department of Psychiatry, Faculty of Medicine, Victor Babes University of Medicine and Pharmacy of Timisoara, 300041 Timisoara, Romania; enatescu.virgil@umft.ro

**Keywords:** neuroinflammation, depression, cytokines, microglia, mood disorders, chronic stress

## Abstract

The relationship between neuroinflammation and depression is a complex area of research that has garnered significant attention in recent years. Neuroinflammation, characterized by the activation of glial cells and the release of pro-inflammatory cytokines, has been implicated in the pathophysiology of depression. The relationship between neuroinflammation and depression is bidirectional; not only can inflammation contribute to the onset of depressive symptoms, but depression itself can also exacerbate inflammatory responses, creating a vicious cycle that complicates treatment and recovery. The present comprehensive review aimed to explore the current findings on the interplay between neuroinflammation and depression, as well as the mechanisms, risk factors, and therapeutic implications. The mechanisms by which neuroinflammation induces depressive-like behaviors are diverse. Neuroinflammation can increase pro-inflammatory cytokines, activate the hypothalamus–pituitary–adrenal (HPA) axis, and impair serotonin synthesis, all of which contribute to depressive symptoms. Furthermore, the activation of microglia has been linked to the release of inflammatory mediators that can disrupt neuronal function and contribute to mood disorders. Stress-induced neuroinflammatory responses can lead to the release of pro-inflammatory cytokines that not only affect brain function but also influence behavior and mood. Understanding these mechanisms is crucial for developing targeted therapies that can mitigate the effects of neuroinflammation on mood disorders.

## 1. Introduction

Depression is a multifaceted mental health disorder characterized by persistent feelings of sadness, hopelessness, and a lack of interest or pleasure in daily activities. It affects millions of individuals worldwide, with significant implications for personal well-being and societal functioning. The World Health Organization (WHO) estimates that depression is a leading cause of disability globally, affecting not only the individual but also families, communities, and economies [1]. The complexity of depression is underscored by its heterogeneous nature, which can manifest in various forms, including major depressive disorder (MDD), persistent depressive disorder (dysthymia), and bipolar disorder, each with distinct clinical features and treatment responses [2]. The prevalence of depression is particularly concerning in vulnerable populations, such as those with chronic illnesses, where the interplay between physical and mental health can exacerbate symptoms and complicate treatment [3]. Neuroinflammation is not only a response to injury or infection but also plays a significant role in the development of various neuropsychiatric disorders, including depression [4]. It has been stated that neuroinflammatory processes can lead to changes in brain structure and function, particularly in regions associated with mood regulation, such as the amygdala and hippocampus [5]. For instance, Zhang et al. highlighted that neuroinflammation in the amygdala correlates with depressive symptoms, suggesting a direct link between inflammatory processes and mood disorders [1]. The importance of neuroinflammation in depression is further supported by findings that anti-inflammatory treatments can yield antidepressant effects, indicating that targeting neuroinflammatory pathways may offer new therapeutic avenues for managing depression [6,7].

Understanding the link between neuroinflammation and depression is of crucial significance for several reasons. First, elucidating this relationship may provide insights into the underlying mechanisms of depression, which could lead to more effective and targeted treatments. Furthermore, the recognition of neuroinflammation as a contributing factor to depression may help in identifying individuals at risk and developing preventive strategies. This is particularly relevant in populations with chronic inflammatory conditions, where the prevalence of depression is notably higher [8]. Additionally, the integration of neuroinflammatory markers into clinical practice could enhance diagnostic accuracy and inform treatment decisions, ultimately improving patient outcomes [9].

The significance of this research extends beyond academic inquiry, as it holds the potential to transform clinical practice and enhance the lives of those grappling with the burdens of depression. The present review aims to elucidate the intricate relationship between neuroinflammatory processes and the pathophysiology of depression. The article systematically examines the multifaceted role of psychological stress as a precursor to neuroinflammation and its subsequent impact on mood disorders. It also delves into the neuroinflammation mechanism in depression, highlighting the relationship between the immune system and depression, the role of microglia in neuroinflammation, and the impact of cytokines on mood disorders. The review also identifies various risk factors contributing to neuroinflammation in depressive disorders and discusses potential therapeutic implications, emphasizing the importance of targeting neuroinflammatory pathways in the treatment of depression. By integrating current research findings, this article seeks to provide a comprehensive understanding of how neuroinflammation may serve as a primary factor in the onset and progression of depression.

## 2. The Role of Biological and Psychological Factors in Depression

### 2.1. Role of Biological Sex

The role of biological sex in depression encompasses genetic, hormonal, and psychosocial factors. Women are approximately twice as likely as men to experience depression, which has been attributed to a variety of biological and psychosocial influences [10]. Hormonal fluctuations, particularly those related to estrogen and progesterone, have been shown to affect mood and stress responses, contributing to the higher prevalence of depression in women [11,12]. Additionally, the interaction between stress hormones, such as cortisol, and biological sex has been highlighted as a significant factor in understanding sex differences in depression. Elevated cortisol levels, often associated with stress, have been linked to depressive symptoms, and this relationship may vary between sexes due to differences in stress perception and response [13,14]. Moreover, sex-specific neurobiological mechanisms play a critical role in the development of mood disorders. Studies have shown that there are distinct differences in the expression of mood-related genes between sexes, which may influence susceptibility to depression [15,16]. Variations in neurotransmitter systems, such as serotonin and dopamine, have been observed to differ between males and females, potentially leading to different manifestations of depressive symptoms [17]. Furthermore, the impact of psychosocial factors, such as societal expectations and gender roles, cannot be overlooked. Women often face unique stressors, including those related to reproductive events and caregiving responsibilities, which can exacerbate their risk of developing depression [18,19].

### 2.2. Types of Psychological Stressors

Psychological stressors are critical elements in understanding the etiology and progression of depression. They can be classified into various types, each with distinct characteristics and implications for mental health. One of the primary classifications of psychological stressors is based on their source, which can be categorized into interpersonal, environmental, and intrapersonal stressors. Interpersonal stressors arise from relationships with others, including conflicts, social isolation, and perceived rejection. Research indicates that interpersonal stressors are significantly correlated with psychological well-being, suggesting that negative social interactions can exacerbate depressive symptoms. For instance, the study by Hay and Diehl found that interpersonal stressors, such as conflicts with family or friends, were strongly associated with increased psychological distress, highlighting the importance of social support in mitigating these effects [20].

Environmental stressors encompass a broader range of external factors, including socioeconomic challenges, exposure to violence, and natural disasters. These stressors can create a chronic state of anxiety and helplessness, which are known risk factors for depression. For example, during the Severe acute respiratory syndrome (SARS) epidemic, college students in China reported significant psychological distress due to stressors related to the outbreak, including fear of illness and disruptions to daily life [21]. Such environmental stressors can lead to prolonged exposure to stress, which has been linked to neurobiological changes associated with depression [22,23].

Intrapersonal stressors, on the other hand, are derived from within the individual and include self-criticism, perfectionism, and chronic worry. These stressors can lead to maladaptive coping mechanisms and contribute to the development of depressive symptoms. For instance, individuals with Type D personality traits, characterized by high levels of negative affectivity and social inhibition, may experience heightened reactivity to psychological stressors, resulting in increased vulnerability to depression. This highlights the interplay between personality traits and stress reactivity, which can exacerbate the effects of psychological stressors on mental health [24].

Another important distinction in the classification of psychological stressors is between acute and chronic stressors. Acute stressors are typically short-term events that can provoke immediate stress responses, such as a job loss or a breakup. While these stressors can lead to temporary distress, they may also serve as catalysts for personal growth and resilience if managed effectively [25]. Conversely, chronic stressors, such as ongoing financial difficulties or long-term caregiving responsibilities, can lead to sustained psychological strain and are more strongly associated with the onset of depression [22]. Chronic exposure to stress can result in dysregulation of the hypothalamic–pituitary–adrenal (HPA) axis, leading to alterations in cortisol levels and other neurobiological changes that predispose individuals to depression [26]. Moreover, the context in which stressors occur can significantly influence their impact on mental health. For example, Montoro and Ceballo found that cultural stressors (e.g., racial discrimination), could compound the effects of other stressors, leading to worse psychological outcomes among Latin adolescents [27].

The role of stigma-related stressors is particularly relevant for marginalized populations, including lesbian, gay, bisexual, transgender, queer or questioning (LGBTQ) individuals. Research by Denton et al. highlighted how perceptions of discrimination can lead to proximal stressors, such as expectations of rejection and internalized stigma, which further exacerbate mental health issues [28]. These stigma-related stressors can create a vicious cycle, where the stress of discrimination leads to increased vulnerability to depression, which in turn can affect social functioning and perpetuate feelings of isolation.

Workplace stressors also represent a significant category of psychological stressors, particularly in high-demand occupations. Studies have shown that job-related stressors, such as job insecurity and high workload, can lead to burnout and depressive symptoms among employees [29]. For instance, humanitarian aid workers often face extreme stress due to the nature of their work, which can lead to significant mental health challenges, including depression. The interplay between workplace stressors and individual coping mechanisms is crucial in understanding how these factors contribute to mental health outcomes [30].

Figure 1 clarifies the types of psychological stressors that can influence neuroinflammation and lead to depression. It not only provides an organized view of how stress interacts with physiological responses but also underscores the complexity of mental health challenges.

### 2.3. Chronic Stress and HPA Axis Dysregulation

Chronic stress is a significant contributor to the development and exacerbation of depression, primarily through its impact on the hypothalamic–pituitary–adrenal (HPA) axis. The HPA axis is a complex neuroendocrine system that plays a crucial role in the body’s response to stress. It involves the hypothalamus, which releases corticotropin-releasing hormone (CRH), stimulating the pituitary gland to secrete adrenocorticotropic hormone (ACTH). This hormone, in turn, prompts the adrenal glands to produce glucocorticoids, primarily cortisol. Under normal circumstances, this system operates through a feedback loop that regulates stress responses. However, chronic stress can lead to dysregulation of this axis, resulting in altered cortisol levels and contributing to the pathophysiology of depression [31]. Prolonged exposure to stressors can lead to sustained activation of the HPA axis, resulting in elevated levels of glucocorticoids [31]. This elevation in glucocorticoids can have detrimental effects on various physiological systems, including the brain, where it can impair neurogenesis and synaptic plasticity, both of which are critical for mood regulation and cognitive function [32]. The glucocorticoid cascade hypothesis posits that chronic overproduction of glucocorticoids can damage glucocorticoid receptors (GRs), particularly in the hippocampus, leading to reduced negative feedback on the HPA axis and perpetuating a cycle of stress and depression [33]. This mechanism can be understood through several interrelated processes involving receptor dynamics, cellular signaling, and neurobiological effects. Zhu et al. demonstrated that glucocorticoids could induce excitotoxicity through increased extracellular glutamate levels, leading to neuronal damage in the hippocampus [34]. This excitotoxicity is exacerbated by the binding of glucocorticoids to GRs, which, when chronically activated, can impair their function and reduce their expression. The downregulation of GRs is significant because it diminishes the receptor’s ability to mediate negative feedback on the HPA axis, resulting in sustained hyperactivity of this stress response system [35]. The interaction between glucocorticoids and mineralocorticoid receptors (MRs) also plays a role in this cascade. Glucocorticoids have a higher affinity for MRs than for GRs, and chronic glucocorticoid excess can lead to MR activation, which may further contribute to hippocampal damage and dysregulation of the HPA axis [36,37]. This dual receptor involvement complicates the physiological stress response, as both GR and MR signaling can influence neuronal health and the overall stress response. The signaling pathways activated by GRs are crucial for mediating the effects of glucocorticoids on gene expression and neuronal function. GRs, upon binding glucocorticoids, translocate to the nucleus and regulate the transcription of target genes [38]. However, chronic glucocorticoid exposure can lead to GR resistance, where the receptor’s ability to activate gene transcription is impaired [38,39]. This resistance can fail to adequately respond to stress, further perpetuating the cycle of stress and depression. Additionally, the neuroprotective mechanisms typically mediated by GRs, such as the modulation of brain-derived neurotrophic factor (BDNF) signaling, are compromised under conditions of chronic glucocorticoid exposure [40,41]. BDNF is essential for neuronal survival and plasticity, and its dysregulation can lead to cognitive deficits and mood disorders commonly associated with chronic stress [41].

The dysregulation of the HPA axis is not merely a consequence of chronic stress; it can also serve as a risk factor for the development of depressive disorders [33]. Additionally, the interaction between chronic stress and HPA axis functioning can vary based on individual differences, such as genetic predispositions and early life experiences, which can influence how one responds to stress [42].

The role of inflammatory cytokines in the context of chronic stress and HPA axis dysregulation has also garnered attention. Research indicates that pro-inflammatory cytokines, such as interleukin-6 (IL-6), can sensitize the HPA axis, leading to exaggerated responses to stress and therefore to the development of depressive symptoms [32]. Furthermore, the circadian regulation of the HPA axis is often disrupted in individuals experiencing chronic stress. This disruption can lead to altered cortisol secretion patterns, such as elevated morning cortisol levels, which have been associated with an increased risk for depression [43]. Schuler et al. found that diurnal cortisol patterns interacted with stressful life events to predict depressive symptoms in adolescent girls, underscoring the importance of considering both stress exposure and HPA axis functioning in understanding depression [44]. The interplay between chronic stress, HPA axis dysregulation, and circadian rhythms highlights the complexity of the biological mechanisms underlying depression.

Chronic dysregulation can lead to long-term changes in brain structure and function, particularly in areas involved in mood regulation, such as the prefrontal cortex and hippocampus [42]. It has been stated that chronic stress has been shown to reduce hippocampal volume, which is associated with memory deficits and increased vulnerability to depression [35]. Therefore, this neurobiological perspective emphasizes the need for interventions targeting HPA axis functioning as a potential therapeutic strategy for individuals suffering from depression.

## 3. Neuroinflammation Mechanisms in Depression

### 3.1. The Immune System and Its Relationship with Depression

The immune system plays an essential role in the pathophysiology of depression, particularly through mechanisms of neuroinflammation. Neuroinflammation refers to the inflammatory response within the central nervous system (CNS), characterized by the activation of glial cells, particularly microglia, and the release of pro-inflammatory cytokines. It has been stated that immune activation is causally linked to depression [45]. Dahl et al. demonstrated that recovery from major depressive disorder (MDD) following non-pharmacological treatment was associated with normalized cytokine levels, suggesting that immune dysregulation plays a significant role in the onset and persistence of depressive symptoms [45]. The role of specific immune cells, such as T helper 17 (Th17) cells, has also been implicated in the relationship between depression and neuroinflammation, through the involvement of Th17 cells in multiple sclerosis (MS) and their potential contribution to depression through neuroinflammatory processes [46]. Th17 cells produce pro-inflammatory cytokines, which can exacerbate neuroinflammation and potentially influence mood disorders. This relationship suggests that immune-mediated mechanisms may play a dual role in both neurodegenerative diseases and psychiatric conditions, further complicating the understanding of depression’s etiology.

The systems genomics approach has provided additional insights into the immune and inflammatory hypotheses of depression. The study by Sharma noted that meta-analyses of genomic, transcriptomic, and proteomic studies consistently implicate immune response pathways in the pathophysiology of depression [47]. Pathway enrichment analyses have revealed that a significant proportion of genes associated with MDD are related to inflammatory responses, suggesting that immune dysregulation may be a fundamental aspect of the disorder. Specific genes such as Dvl3 have been identified as influencing MDD susceptibility through their interactions with inflammatory pathways. Dvl3 polymorphisms have been shown to correlate with pro-inflammatory cytokine levels, indicating a genetic predisposition to inflammatory responses in MDD [48]. Additional key genes that have been associated with inflammatory responses in MDD include those involved in the MAPK signaling pathway, such as p38 MAPK, ERK, B-Raf, and MAPK1, which have been identified as downstream targets of this pathway, playing significant potential roles in the pathogenesis of MDD [49,50,51]. Moreover, a broader set of differentially expressed genes (DEGs) in MDD have been identified that are enriched for pathways related to inflammation and immune responses. For example, Gialluisi et al. identified a set of 165 DEGs, with a significant number linked to inflammatory processes, further supporting the notion that immune dysregulation is a fundamental aspect of MDD [52].

Neuroinflammation is often triggered by stress, which can activate microglia, the resident immune cells of the CNS. Chronic stress has been shown to induce a pro-inflammatory state in microglia, leading to the release of cytokines such as interleukin-1β (IL-1β) and tumor necrosis factor-alpha (TNF-α) [53]. These cytokines can disrupt neurotransmitter systems and neuroplasticity, contributing to the development of depressive symptoms. For example, Chen and Cao demonstrated that neuroinflammation in lipopolysaccharide (LPS)--stimulated microglial cells can lead to neuronal apoptosis, highlighting the detrimental effects of activated microglia on neuronal health and function [54]. Moreover, the interplay between neuroinflammation and the kynurenine pathway has garnered attention in the context of depression. Troubat et al. proposed a revised serotonin-kynurenine-inflammation hypothesis of depression, suggesting that increased levels of glucocorticoids and systemic inflammation can shift tryptophan metabolism towards the kynurenine pathway, thereby reducing serotonin availability and contributing to depressive symptoms [55]

The role of microglia in depression has been studied, indicating that their activation can lead to neuroinflammation and subsequent behavioral changes. Jia et al. emphasized that microglial abnormalities are implicated in a range of neuropsychiatric disorders, including depression [56]. Activated microglia release pro-inflammatory cytokines that can disrupt neuronal signaling and contribute to the development of depressive-like behaviors [56]. Furthermore, the relationship between neuroinflammation and depression is not limited to peripheral immune responses; central inflammation can also play a critical role. The neuroinflammatory processes occurring within the CNS can directly influence mood and behavior, reinforcing the idea that depression is closely linked to immune system dysregulation [57]. In addition to cytokines, other factors such as the integrity of the blood–brain barrier (BBB) are crucial in mediating the relationship between the immune system and depression. Benatti et al. noted that altered BBB permeability could facilitate the infiltration of peripheral immune cells into the CNS, exacerbating neuroinflammation and contributing to depressive symptoms [58].

The influence of gut microbiota on neuroinflammation and depression has also emerged as a significant area of research. Yao et al. demonstrated that gut microbiota can regulate neuroinflammation through mechanisms involving the *NLRP3* inflammasome, which is implicated in the development of depressive-like behaviors. This connection between the gut microbiome, immune responses, and neuroinflammation underscores the complexity of the interactions between various biological systems in the context of depression [59]. Figure 2 presents a complex diagram illustrating the gut–brain–immune axis and its involvement in neuroinflammatory processes, particularly through Th17 cells and their impact on neuroinflammation and mood disorders. Th17 cells originating from the gut infiltrate the CNS, releasing cytokines, which activate neurons, astrocytes, and microglia, leading to neuroinflammation. This inflammation is linked to mood disturbances and stress, which may further affect gut health, creating a feedback loop between the gut and the brain.

### 3.2. The Role of Microglia in Neuroinflammation

Microglia, the resident immune cells of the CNS, play a crucial role in neuroinflammation and have emerged as key players in the pathophysiology of depression. These cells are responsible for maintaining homeostasis in the brain, responding to injury, and modulating immune responses. However, their activation can lead to neuroinflammatory processes that contribute to the development and exacerbation of depressive symptoms. Under normal physiological conditions, microglia exhibit a ramified morphology, characterized by long, branched processes that allow them to continuously survey the brain environment [60]. In response to various stimuli, such as injury or infection, microglia can become activated, transitioning to an amoeboid shape and proliferating in the affected area. This activation is typically accompanied by the release of pro-inflammatory cytokines, including interleukin-1β (IL-1β), tumor necrosis factor-alpha (TNF-α), and interleukin-6 (IL-6), which can further propagate neuroinflammation and contribute to neuronal damage. The dysregulation of this process is particularly relevant in the context of depression, where chronic neuroinflammation is thought to play a significant role in the onset and persistence of depressive symptoms [61,62].

The dual role of microglia in neuroinflammation, where they can exhibit both pro-inflammatory (M1) and anti-inflammatory (M2) phenotypes, was highlighted [63,64]. The M1 phenotype is associated with the release of inflammatory mediators that can exacerbate neuronal injury, while the M2 phenotype is involved in tissue repair and resolution of inflammation. The balance between these two phenotypes is critical; an overactive M1 response can lead to sustained neuroinflammation, contributing to the pathogenesis of depression [65]. Wohleb et al. demonstrated that stress-induced microglial activation leads to a shift towards the M1 phenotype, which is associated with depressive-like behaviors in animal models. This suggests that targeting microglial polarization may offer therapeutic potential in treating depression [66].

Figure 3 presents the two phenotypes expressed during microglial activation. In a dichotomous framework, microglial activation is contingent upon the nature of the stimuli, leading to the emergence of distinct phenotypes. Microglia exhibiting an M1 phenotype are characterized by the expression of pro-inflammatory cytokines, chemokines, and neurotoxic factors, whereas M2-polarized microglia predominantly secrete anti-inflammatory, neuroprotective, and wound-healing mediators. The activation and polarization of microglia occur under both resting conditions and during episodes of neuroinflammation. Under physiological conditions, patrolling microglia plays a crucial role in maintaining CNS homeostasis. Conversely, during neuroinflammatory responses, microglia adopt an amoeboid morphology and transition to either the classical M1 or alternative M2 phenotype, influenced by the characteristics of the local microenvironment.

Microglia also play a significant role in synaptic pruning, a process essential for normal brain development and function. However, excessive synaptic pruning mediated by activated microglia can lead to synaptic loss and contribute to cognitive deficits associated with depression [67]. It has been stated that microglia can engulf synapses during periods of heightened inflammation, leading to impaired synaptic connectivity and function [68].

The interaction of microglia with other cell types in the CNS, such as astrocytes and neurons, further complicates the role of microglia in neuroinflammation. Activated microglia can release signaling molecules that influence astrocytic function, exacerbating neuroinflammation [69]. Additionally, microglia can respond to neuronal signals, such as exosomes containing microRNAs, which can promote microglial activation and the subsequent inflammatory response [70]. The mechanisms underlying microglial activation in depression are multifactorial. For instance, Dong et al. reported that lipopolysaccharide (LPS) exposure, a model of systemic inflammation, leads to significant microglial activation and the release of pro-inflammatory cytokines, which can contribute to depressive-like behaviors [71]. This suggests that chronic exposure to stressors may prime microglia for an exaggerated inflammatory response, further perpetuating the cycle of neuroinflammation and depression. Moreover, genetic and epigenetic factors may also influence microglial function and their role in neuroinflammation [69]. Variants in genes associated with immune responses, such as *TREM2* (triggering receptor expressed on myeloid cells 2) and CD33, have been linked to an increased risk of neurodegenerative diseases like Alzheimer’s disease (AD) and related disorders. These genetic predispositions may affect microglial activation and the subsequent inflammatory response, highlighting the need for personalized approaches to treatment based on individual genetic profiles [69]. Mutations in *TREM2*, particularly the *R47H* variant, have been shown to significantly increase the risk of developing AD and other neurodegenerative conditions [72,73]. This receptor is primarily expressed in microglia and is crucial for their activation and function, particularly in phagocytosis—the process by which microglia clear amyloid-beta plaques associated with AD [74,75]. Loss-of-function mutations in *TREM2* lead to impaired microglial function, resulting in reduced clearance of neurotoxic agents and increased neuroinflammation, which contributes to neurodegeneration [76,77]. Furthermore, *TREM2*’s interaction with apolipoproteins such as APOE enhances its ability to facilitate the uptake of amyloid-beta, thereby linking genetic variations in *TREM2* to the pathology of AD [73,74]. Contrarily, the CD33 gene encodes a sialic acid-binding immunoglobulin-like lectin expressed on myeloid cells, including microglia. Genetic variants of CD33 have been implicated in modulating the phagocytic activity of microglia, with certain alleles associated with decreased uptake of amyloid-beta [78,79]. This inhibition of phagocytosis by CD33 variants can lead to an accumulation of amyloid plaques, exacerbating neuroinflammatory processes and contributing to AD pathology [80,81]. The interplay between CD33 and *TREM2* is particularly noteworthy; while *TREM2* enhances microglial phagocytosis, CD33 appears to inhibit it, suggesting a complex regulatory network that influences the immune response in the brain [80,82]. In addition, the variants in *TREM2* and CD33 may contribute to immune dysregulation, which can affect mood and cognitive functions [83,84], as their roles in modulating microglial activity can influence the inflammatory milieu of the brain, potentially leading to depressive symptoms [73,85]. Therefore, the chronic activation of microglia and the resultant inflammatory cytokine release can disrupt neural circuits involved in mood regulation, linking immune response genes to both neurodegenerative diseases and mood disorders [77,86].

Therapeutically, targeting microglial activation and modulating their inflammatory responses may offer new avenues for treating depression. For instance, anti-inflammatory agents, such as interleukin-10 (IL-10), have shown promise in reducing microglial activation and ameliorating depressive symptoms in preclinical models [87]. Additionally, compounds that promote M2 polarization of microglia may enhance their neuroprotective functions and mitigate the detrimental effects of neuroinflammation [88]. These strategies underscore the potential for developing novel treatments that address the underlying neuroinflammatory mechanisms associated with depression.

### 3.3. Cytokines and Their Impact on Mood Disorders

Cytokines are small signaling proteins that play a significant role in mediating immune responses and have been increasingly recognized for their impact on mood disorders, particularly depression. These molecules are produced by various immune cells, including microglia, macrophages, and lymphocytes, and they can influence neuronal function and behavior through their pro-inflammatory and anti-inflammatory properties. One of the most well-studied pro-inflammatory cytokines in depression is interleukin-6 (IL-6). Elevated levels of IL-6 have been associated with depressive symptoms and are thought to contribute to the pathophysiology of mood disorders [89]. This finding is consistent with the inflammation theory of depression, which posits that chronic inflammation may lead to alterations in neurotransmitter systems, neuroplasticity, and, ultimately, mood dysregulation [90,91].

Another key cytokine implicated in mood disorders is tumor necrosis factor-alpha (TNF-α). TNF-α is produced by activated microglia and is known to have neurotoxic effects when present in elevated concentrations. The TNF-α can disrupt the function of the HPA axis, leading to increased cortisol levels and glucocorticoid resistance. The dysregulation of the HPA axis due to pro-inflammatory cytokines like TNF-α may contribute to the development of depressive symptoms, particularly in individuals with chronic stress or inflammatory conditions [92]. In addition, cytokines can also influence neurogenesis, a process that is often impaired in individuals with depression. Exposure to IL-1β inhibited adult hippocampal neurogenesis, which is critical for mood regulation and cognitive function [93].

The relationship between cytokines and mood disorders is not limited to pro-inflammatory cytokines; anti-inflammatory cytokines also play a significant role. For instance, interleukin-10 (IL-10) is known for its anti-inflammatory properties and has been shown to counteract the effects of pro-inflammatory cytokines. Lower levels of IL-10 were associated with mood disturbances in bipolar disorder, indicating that a deficiency in anti-inflammatory signaling may contribute to the pathogenesis of mood disorders [87]. Pro-inflammatory cytokines such as IL-6 and TNF-α contribute to the dysregulation of neurotransmitter systems, impair neurogenesis, and disrupt the HPA axis, while anti-inflammatory cytokines like IL-10 may help mitigate these effects. Moreover, the activation of the immune system and the subsequent release of cytokines can perpetuate neuroinflammation [94,95].

### 3.4. HPA Axis Activation and Inflammation

Figure 4 illustrates the intricate relationship within the HPA axis. In response to a physiological or psychological stressor, the hypothalamus initiates the stress response by secreting corticotropin-releasing factor (CRH), a peptide hormone crucial for activating the HPA axis. Upon release, CRH acts on the anterior pituitary gland, stimulating it to produce and secrete adrenocorticotropic hormone (ACTH). ACTH, in turn, enters the systemic circulation and targets the adrenal cortex, where it prompts the production and release of cortisol, a primary glucocorticoid hormone. Cortisol plays a key role in the body’s stress response, facilitating energy mobilization, modulating immune responses, and influencing mood and behavior. Importantly, cortisol also exerts negative feedback on both the anterior pituitary gland and the hypothalamus, thereby regulating its production and maintaining homeostasis within the system. This feedback mechanism ensures that cortisol levels do not become excessively elevated, which could lead to detrimental effects on health [96,97].

Research has shown that individuals with MDD often exhibit elevated levels of cortisol, which is associated with increased HPA axis activity [96]. This hyperactivity can lead to a state of glucocorticoid resistance, where the body’s tissues become less responsive to the regulatory effects of cortisol, further perpetuating the cycle of stress and depression [98].

The interplay between HPA axis activation and inflammation is complex and bidirectional. On one hand, chronic stress and the resulting HPA axis activation can lead to increased production of pro-inflammatory cytokines, such as IL-6 and TNF-α, which are known to contribute to neuroinflammation [92]. This neuroinflammation can disrupt neurotransmitter systems, impair neuroplasticity, and ultimately contribute to the development of depressive symptoms [99]. Elevated levels of IL-6 have been associated with depressive symptoms, suggesting that inflammation may mediate the effects of HPA axis dysregulation on mood [100]. On the other hand, neuroinflammation itself can influence HPA axis activity. Cytokines released during inflammatory responses can act on the hypothalamus, promoting the release of CRH and further stimulating the HPA axis [101,102].

## 4. Risk Factors for Neuroinflammation in Depression

### 4.1. Genetic Factors

Genetic factors play a significant role in the susceptibility to neuroinflammation and its subsequent impact on depression. The interplay between genetic predispositions and environmental stressors can influence inflammatory responses in the brain, contributing to the development and persistence of mood disorders. One of the key genetic factors implicated in the relationship between inflammation and depression is the polymorphism of genes involved in the inflammatory response. For instance, the study reported by Barnes et al. highlighted that the ‘at risk’ G/G genotype of the IL-6 gene was associated with a higher severity of somatic symptoms of depression in patients undergoing interferon-alpha (IFN-α) treatment [103]. This suggests that individuals with certain genetic variants may have an increased vulnerability to the effects of inflammatory cytokines, leading to a higher risk of developing depressive symptoms. Moreover, the relationship between genetic factors and inflammation is also supported by research examining the role of C-reactive protein (CRP) and interleukin-6 (IL-6) in depression. Haapakoski et al. conducted a cumulative meta-analysis that demonstrated elevated levels of IL-6 and CRP in patients with major depressive disorder, suggesting a strong association between inflammation and depression [104]. Genetic factors may contribute to this association, as certain polymorphisms in inflammatory genes can influence the production and regulation of these cytokines. For example, variations in the gene encoding for IL-1β have been linked to increased levels of this cytokine in individuals with depression, highlighting the role of genetic predisposition in modulating inflammatory responses.

The genetic basis for neuroinflammation in depression is also supported by studies examining the relationship between chronic low-grade inflammation and mood disorders [105]. Khandaker et al. found that while there was no direct association between genetic risk factors for coronary heart disease and depression, shared genetic mechanisms may influence both conditions [106]. This indicates that genetic predispositions may not only affect susceptibility to depression but also modulate the inflammatory processes that contribute to its pathophysiology. Furthermore, the role of the indoleamine-2,3-dioxygenase (IDO) gene has been implicated in the relationship between inflammation and depression. IDO is an enzyme that catalyzes the conversion of tryptophan to kynurenine, which can be upregulated during inflammatory responses. Elevated levels of kynurenine have been associated with depressive symptoms, and genetic polymorphisms in the IDO gene may influence the activity of this enzyme [107,108,109]. In addition to specific gene polymorphisms, epigenetic factors also play a role in the relationship between inflammation and depression. Environmental stressors can lead to epigenetic modifications that influence gene expression, potentially affecting inflammatory responses in the brain [110]. For example, exposure to chronic stress can lead to DNA methylation changes in genes associated with inflammation, which may contribute to the development of depressive symptoms [111]. Several genes that are significantly affected by stress-related DNA methylation changes have been identified in the literature, leading to alterations in their expression and function. One prominent gene is the serotonin transporter gene (*SLC6A4*). An increased methylation of the *SLC6A4* promoter is associated with higher levels of depressive symptoms. This gene plays a crucial role in serotonin uptake, and its downregulation due to methylation can lead to decreased serotonin activity, which is a well-established factor in the pathophysiology of depression [112,113]. Additionally, the *FKBP5* gene, which is involved in the regulation of the stress response, has also been implicated. Changes in its methylation status have been linked to stress exposure and depressive symptoms, suggesting that *FKBP5* may mediate the effects of chronic stress on mood disorders [114,115]. Another critical gene is the brain-derived neurotrophic factor (BDNF), which is essential for neuronal survival and plasticity. Chronic stress has been associated with decreased BDNF expression, potentially mediated by DNA methylation changes [116,117]. Furthermore, the gene encoding the glucocorticoid receptor *(NR3C1*) has been shown to undergo increased methylation in response to stress, leading to impaired stress response mechanisms and heightened vulnerability to depression [118,119].

The implications of genetic factors in neuroinflammation and depression extend to treatment strategies as well. Raison et al. conducted a randomized controlled trial of the tumor necrosis factor (TNF) antagonist infliximab in patients with treatment-resistant depression, demonstrating that those with elevated inflammatory markers had a better response to treatment [120]. This suggests that personalized treatment approaches based on genetic and inflammatory profiles may enhance the efficacy of interventions for depression.

### 4.2. Environmental Stressors

Environmental stressors are significant contributors to the risk of neuroinflammation in depression, influencing both peripheral and central immune responses. These stressors can include a wide range of factors such as pollution, psychosocial stress, dietary influences, and physical injuries.

One of the primary mechanisms by which environmental stressors induce neuroinflammation is through systemic immune activation. For example, exposure to air pollution has been shown to increase levels of pro-inflammatory cytokines in the bloodstream, which can subsequently cross the blood–brain barrier and activate microglia in the brain. Bolton et al. demonstrated that maternal stress and prenatal exposure to air pollution could have long-lasting effects on offspring, leading to altered microglial function and increased neuroinflammation, which are associated with anxiety and depressive behaviors [121]. Another significant environmental stressor is oxidative stress, which can result from various factors, including exposure to pollutants, poor diet, and chronic psychological stress. Oxidative stress can lead to cellular damage and inflammation, contributing to the dysregulation of neurotransmitter systems involved in mood regulation. For instance, Tyler et al. found that aging exacerbates neuroinflammatory outcomes induced by acute ozone exposure, suggesting that environmental pollutants can have a cumulative effect on brain health and function [122].

The impact of environmental stressors on neuroinflammation is further supported by studies examining the effects of psychosocial stress [123,124]. Research by Fagundes et al. demonstrated that depressive symptoms could enhance stress-induced inflammatory responses, indicating a bidirectional relationship between psychological stress and inflammation [125]. Additionally, the role of the sympathetic nervous system (SNS) in mediating the effects of environmental stressors on neuroinflammation is noteworthy. Stress-induced activation of the SNS can lead to the release of catecholamines, which can influence immune responses and promote inflammation. Niraula et al. found that corticosterone production during repeated social defeat stress caused monocyte mobilization from the bone marrow, leading to increased neuroinflammation and prolonged anxiety-like behavior [126].

The cumulative effects of environmental stressors on neuroinflammation can also be observed in the context of neurodegenerative diseases. It has been stated that chronic exposure to environmental toxins, such as pesticides and heavy metals, can lead to increased neuroinflammation and contribute to the development of mood disorders [127]. Moreover, the role of genetic factors in modulating the effects of environmental stressors on neuroinflammation is an important consideration. Individuals with specific genetic polymorphisms may be more susceptible to the inflammatory effects of environmental stressors, increasing their risk of developing depression [128,129]. For example, variations in genes associated with the immune response, such as those encoding for cytokines or their receptors, can influence how individuals respond to environmental stressors and their subsequent risk for mood disorders [130]. Several key genes and their implications in mood disorders have been identified through various studies. One example is the IL-1β gene. Polymorphisms in the IL-1β gene can lead to increased expression of IL-1β, which has been shown to influence gut microbiota balance and mood status following psychophysiological stress [131]. Elevated levels of IL-1β are associated with inflammatory disorders, which can act as a cofactor in the onset of mood disorders. Additionally, the TNF-α gene is another critical player. TNF-α is recognized as a neuromodulator in the brain and has been implicated in the pathophysiology of mood disorders, with evidence suggesting that its signaling pathways may serve as potential targets for antidepressant drug development [87,132]. Moreover, the glucocorticoid receptor (*NR3C1*) gene is essential in regulating the body’s response to stress. Variations in this gene can disrupt glucocorticoid regulation, leading to heightened inflammation and increased susceptibility to mood disorders [133]. The inducible nitric oxide synthase (iNOS) gene is also noteworthy, as it is involved in the production of reactive nitrogen species and has been linked to immune responses during stress [134]. Furthermore, the IFN-γ gene, a critical cytokine in immune response, has been shown to interact with stress-related pathways, influencing the development of mood disorders [135].

### 4.3. Lifestyle Factors

Lifestyle factors, including diet, beverages enriched with anti-inflammatory molecules, exercise, and sleep, play a significant role in the risk of neuroinflammation and the development of depression. These factors can influence the body’s inflammatory responses and neurobiological processes, thereby affecting mood regulation and cognitive function.

Diet is a critical determinant of overall health and has been shown to influence inflammation in the body. A diet rich in anti-inflammatory foods, such as fruits, vegetables, whole grains, and omega-3 fatty acids, has been associated with lower levels of pro-inflammatory cytokines and a reduced risk of depression [136]. Conversely, diets high in processed foods, sugars, and saturated fats can promote inflammation and have been linked to an increased risk of mood disorders [137]. For instance, Rutledge et al. found that poor dietary habits, characterized by low intake of fruits and vegetables and high intake of processed foods, were associated with higher levels of depressive symptoms among women with suspected myocardial ischemia [136]. This suggests that dietary choices can significantly affect neuroinflammatory processes and, consequently, mood regulation. The Mediterranean diet, which emphasizes the consumption of whole foods, healthy fats, and lean proteins, has been shown to have protective effects against depression. Research by García-Toro et al. indicated that adherence to a Mediterranean diet was associated with improved mood and reduced depressive symptoms in individuals with metabolic syndrome [138]. This dietary pattern is believed to exert its beneficial effects through anti-inflammatory mechanisms, promoting the production of neurotrophic factors and supporting overall brain health.

The consumption of beverages enriched with anti-inflammatory molecules has garnered attention for its potential beneficial effects on depressive symptoms. Beverages such as coffee, tea, and those enriched with polyphenols have been shown to possess anti-inflammatory properties. For instance, coffee, one of the most widely consumed beverages globally, contains bioactive compounds that exhibit both anti-inflammatory and antioxidant activities, which may contribute to a reduced risk of developing depression [139]. Similarly, green tea, rich in epigallocatechin gallate (EGCG), has demonstrated significant anti-inflammatory effects that could help mitigate depressive symptoms [140]. The anti-inflammatory effects of these beverages are thought to stem from their ability to lower levels of inflammatory markers such as C-reactive protein (CRP) and IL-6, which are often elevated in individuals with depression [141]. Moreover, the incorporation of fermented beverages, such as those containing probiotics, has shown promise in enhancing gut health and reducing inflammation, which may further alleviate depressive symptoms [142]. The gut–brain axis is a critical pathway through which dietary components can influence mental health, with emerging evidence suggesting that probiotics can modulate inflammatory responses and improve mood [142]. Polyphenol supplementation, particularly through beverages, has garnered attention for its potential beneficial effects on depressive symptoms. The study reported by Pase et al. demonstrated that cocoa polyphenols could enhance positive mood states, although they did not significantly affect cognitive performance [143]. This aligns with findings from Lin et al., which highlighted that polyphenol supplementation could improve depression and anxiety symptoms, particularly in specific populations such as menopausal women [144]. The anxiolytic and antidepressant effects of polyphenols have been corroborated by various studies, indicating that these compounds may influence neurotransmitter systems and neurotrophic factors, such as BDNF, which are crucial for mood regulation [145,146]. Moreover, consuming polyphenol-rich foods and beverages has improved overall mental well-being [147]. A systematic review by Bayes et al. found that numerous studies reported significant effects of polyphenols on depressive symptoms across various populations [148]. This is further supported by the work of Barfoot et al., which indicated that higher levels of specific flavonoids, such as anthocyanins and flavanones, were associated with reduced symptoms of depression in postpartum women [149]. Additionally, polyphenols may enhance cerebral blood flow, which is essential for optimal brain function and mood regulation. Research has shown that improved endothelial function, as a result of polyphenol intake, correlates with better mood states [150].

Regular physical activity is another important lifestyle factor that can influence neuroinflammation and mood. Exercise has been shown to have anti-inflammatory effects, reducing levels of pro-inflammatory cytokines and promoting the release of anti-inflammatory mediators. For example, Ji et al. found that exercise improved sleep quality and reduced anxiety and depressive symptoms in college students, highlighting the mental health benefits of physical activity [151]. Additionally, Takács demonstrated that even a short three-week exercise program could lead to significant improvements in sleep quality and reductions in depressive symptoms, suggesting that exercise can have immediate positive effects on mood [152]. The mechanisms through which exercise exerts its anti-inflammatory effects are multifaceted. Exercise can enhance the production of neurotrophic factors, such as BDNF, which are essential for neurogenesis and synaptic plasticity [153]. Furthermore, physical activity has been shown to improve sleep quality, which is closely linked to mood regulation and inflammation. The interaction between exercise, sleep, and inflammation underscores the importance of maintaining an active lifestyle for mental health [154].

Sleep is a vital component of overall health and well-being, and disturbances in sleep patterns are commonly observed in individuals with depression. Poor sleep quality and sleep disorders can exacerbate neuroinflammation and contribute to the development of mood disorders [138]. Research has shown that sleep deprivation can lead to increased levels of pro-inflammatory cytokines, which can further impair mood and cognitive function [155]. For instance, Passos et al. found that exercise improved sleep quality and reduced inflammatory markers in patients with chronic primary insomnia, suggesting that addressing sleep disturbances may help mitigate neuroinflammation and improve mood [156]. The relationship between sleep and neuroinflammation is complex and bidirectional. While poor sleep can lead to increased inflammation, chronic inflammation can also disrupt sleep patterns, creating a cycle that perpetuates mood disorders [157]. For example, elevated levels of inflammatory cytokines have been associated with sleep disturbances, including insomnia and obstructive sleep apnea, which are prevalent in individuals with depression [158]. This highlights the importance of addressing both sleep quality and inflammation in the management of mood disorders.

### 4.4. Comorbidities

Comorbidities are significant risk factors for neuroinflammation in depression, as they can exacerbate inflammatory responses and contribute to the complexity of mood disorders. The presence of additional physical or mental health conditions alongside depression can complicate diagnosis, treatment, and overall patient outcomes.

Physical health conditions, such as cardiovascular disease, diabetes, and chronic pain, are frequently observed in individuals with depression. These comorbidities can contribute to systemic inflammation, which may, in turn, exacerbate neuroinflammation in the brain. Boudreau et al. found that patients with chronic migraine and comorbid depression exhibited increased levels of inflammatory markers, suggesting that the inflammatory processes associated with migraine may contribute to the development of depressive symptoms [159].

Chronic conditions such as obesity and metabolic syndrome are also associated with increased levels of pro-inflammatory cytokines, which can influence neuroinflammatory processes in the brain. Winkler et al. reported that individuals with depression and comorbid anxiety exhibited higher rates of physical comorbidities, including hypertension and type II diabetes, which are known to be linked to chronic inflammation [160]. The presence of these comorbidities can create a feedback loop, where inflammation from physical health issues exacerbates depressive symptoms, leading to further deterioration of both physical and mental health [161,162].

In addition to physical health conditions, psychological comorbidities such as anxiety disorders and post-traumatic stress disorder (PTSD) are commonly associated with depression. The presence of these comorbid conditions can intensify neuroinflammatory responses and contribute to the severity of depressive symptoms. Fusar-Poli et al. found that individuals with an at-risk mental state who also exhibited comorbid depressive and anxiety disorders experienced greater psychopathology and a higher risk of transitioning to psychosis [163]. This suggests that the interplay between different psychiatric conditions can exacerbate neuroinflammation and mood dysregulation.

The presence of comorbidities can significantly affect treatment outcomes for individuals with depression. Sertöz et al. found that patients with acute myocardial infarction who also had comorbid depression experienced poorer quality of life and worse treatment outcomes compared to those without depression [164]. Moreover, the treatment of comorbid conditions can influence the effectiveness of antidepressant therapies [165]. For example, individuals with osteoarthritis and comorbid depression may respond differently to treatment than those with depression alone. Agarwal et al. reported that nearly 25% of individuals with osteoarthritis and depression also reported anxiety disorders, indicating that the presence of multiple comorbidities can complicate treatment strategies [166].

The presence of physical and psychological health conditions can exacerbate inflammatory responses, leading to increased vulnerability to mood disorders. By addressing both the mental and physical health aspects of comorbid conditions, healthcare providers can enhance the overall well-being of patients and reduce the impact of neuroinflammation on mood regulation.

## 5. Therapeutic Implications of Neuroinflammation in Depression

### 5.1. Biomarkers of Inflammation in Depressed Patients

The identification of biomarkers of inflammation in depressed patients has emerged as an emerging area of research, providing insights into the pathophysiology of depression and potential therapeutic targets. Inflammation is increasingly recognized as a contributing factor to the development and persistence of depressive symptoms, and various biomarkers have been associated with neuroinflammatory processes in mood disorders.

Pro-inflammatory cytokines, such as IL-6, TNF-α, and IL-1β, are among the most studied biomarkers of inflammation in depression. Elevated levels of these cytokines have been consistently observed in individuals with MDD and are associated with increased severity of depressive symptoms [167]. The relationship between pro-inflammatory cytokines and depression is further supported by studies examining the effects of cytokine treatment on mood. Patients receiving IFN-α therapy for chronic viral infections often develop depressive symptoms as a side effect of treatment, which is thought to be mediated by the inflammatory response induced by cytokines [89].

C-reactive protein (CRP) is another important biomarker of inflammation that has been linked to depression [168]. Elevated levels of CRP have been associated with an increased risk of developing depressive symptoms and are thought to reflect systemic inflammation [169]. Individuals with elevated CRP levels are more likely to experience treatment-resistant depression, indicating that inflammation may play a role in the efficacy of antidepressant therapies [170].

Neurotrophic factors, such as BDNF, are critical for neuronal health and function. Levels of BDNF are often reduced in individuals with depression, and this reduction is associated with increased levels of pro-inflammatory cytokines [171]. Isgren et al. found that lower BDNF levels were correlated with higher levels of inflammatory markers in patients with bipolar disorder, suggesting that neuroinflammation may contribute to the dysregulation of neurotrophic factors in mood disorders [170].

The translocator protein (TSPO) has gained attention as a potential biomarker of neuroinflammation in depression. TSPO is expressed in activated microglia and is involved in the inflammatory response in the brain. Studies using positron emission tomography (PET) imaging have shown increased TSPO binding in the brains of individuals with MDD, indicating heightened microglial activation and neuroinflammation [172].

The identification of inflammatory biomarkers in depressed patients has important implications for treatment strategies. Understanding the role of inflammation in depression may lead to the development of novel therapeutic approaches that target inflammatory processes. Anti-inflammatory agents, such as cytokine inhibitors and non-steroidal anti-inflammatory drugs (NSAIDs), have shown promise in alleviating depressive symptoms in individuals with elevated inflammatory markers [173,174,175].

### 5.2. Pharmacological Interventions

The recognition of neuroinflammation as a significant contributor to the pathophysiology of depression has led to increased interest in anti-inflammatory treatments as potential therapeutic strategies.

Several classes of anti-inflammatory medications have been investigated for their efficacy in treating depression. Non-steroidal anti-inflammatory drugs (NSAIDs), corticosteroids, and cytokine inhibitors have shown promise in alleviating depressive symptoms, particularly in individuals with elevated inflammatory markers. A meta-analysis by Köhler-Forsberg et al. demonstrated that anti-inflammatory treatments could significantly reduce depressive symptoms and improve response rates in patients with MDD [167]. The effect sizes observed for these treatments were notably large, indicating that inflammation may be a modifiable target in the management of depression.

One of the most compelling examples of anti-inflammatory treatment in depression is the use of cytokine inhibitors. For instance, the tumor necrosis factor-alpha (TNF-α) inhibitor infliximab has been shown to have antidepressant effects in treatment-resistant depression, particularly in patients with elevated levels of C-reactive protein (CRP). In a randomized controlled trial, patients with high CRP levels who received infliximab demonstrated significant reductions in depressive symptoms compared to those receiving a placebo [176]. This suggests that targeting specific inflammatory pathways may enhance treatment outcomes for individuals with depression.

Additionally, corticosteroids, which are anti-inflammatory agents that mimic the effects of cortisol, have been explored for their potential to alleviate depressive symptoms. While chronic use of corticosteroids can lead to mood disturbances, short-term administration has been shown to have antidepressant effects in some patients. However, the long-term implications of corticosteroid use in depression remain a topic of ongoing research, as prolonged exposure can lead to HPA axis dysregulation and increased risk of mood disorders [104,177].

Emerging therapies targeting neuroinflammation are also being explored as potential treatments for depression. For instance, the use of ketamine, an NMDA receptor antagonist, has shown rapid antidepressant effects in treatment-resistant depression [178]. Ketamine may exert its effects, in part, by reducing neuroinflammation and promoting neuroplasticity.

Another promising area of research involves the use of probiotics and dietary supplements to modulate the gut–brain axis and reduce neuroinflammation. The gut microbiota has been shown to influence inflammatory responses and neurotransmitter production, suggesting that interventions aimed at improving gut health may have beneficial effects on mood [176]. Probiotics can reduce levels of inflammatory markers and improve mood in individuals with depression [179]. This suggests that targeting the gut–brain axis may offer a novel approach to managing neuroinflammation and depression.

### 5.3. Modifiable Lifestyle Factors

The integration of dietary modifications, exercise, and sleep into a comprehensive lifestyle intervention program has shown promise in reducing inflammation and improving mood. A randomized controlled trial by Garcia-Toro et al. demonstrated that a combination of dietary changes, physical activity, sunlight exposure, and improved sleep patterns led to significant reductions in depressive symptoms among patients with major depressive disorder [157]. This suggests that a holistic approach to lifestyle modification can effectively address the underlying neuroinflammatory processes associated with depression. Moreover, the potential for lifestyle modifications to serve as adjunctive treatments for depression is supported by the following studies examining the effects of lifestyle interventions on mental health outcomes.

A diet rich in anti-inflammatory foods, such as fruits, vegetables, whole grains, and omega-3 fatty acids, has been associated with lower levels of pro-inflammatory cytokines and a reduced risk of depression [180]. The Mediterranean diet, characterized by high consumption of plant-based foods and healthy fats, has been linked to improved mood and lower levels of inflammatory markers [181]. Conversely, diets high in processed foods, sugars, and saturated fats can promote systemic inflammation, which may contribute to the development of depressive symptoms [182]. Specific dietary components can influence the production of inflammatory markers. Omega-3 fatty acids, found in fatty fish and flaxseeds, have been shown to exert anti-inflammatory effects and improve mood [182,183]. Additionally, the consumption of dietary fiber has been linked to improved mental health outcomes, potentially through its effects on gut microbiota and subsequent modulation of inflammation [184].

Exercise has been shown to have anti-inflammatory effects, reducing levels of pro-inflammatory cytokines and promoting the release of anti-inflammatory mediators [185]. Hayward et al. found that as little as 10–29 min of daily physical activity may significantly reduce the relative risk of clinical depression in women [186]. The mechanisms through which exercise exerts its anti-inflammatory effects are multifaceted. Exercise can enhance the production of neurotrophic factors, such as BDNF, which are essential for neurogenesis and synaptic plasticity [187]. Furthermore, physical activity has been shown to improve sleep quality, which is closely linked to mood regulation and inflammation [188].

Poor sleep quality and sleep disorders can exacerbate neuroinflammation and contribute to the development of mood disorders [189]. Sleep deprivation can lead to increased levels of pro-inflammatory cytokines, which can further impair mood and cognitive function [190]. The relationship between sleep and inflammation is complex and bidirectional. While poor sleep can lead to increased inflammation, chronic inflammation can also disrupt sleep patterns, creating a cycle that perpetuates mood disorders [181].

Overall, diet and lifestyle factors play a significant role in modulating inflammation and influencing the risk of depression. A healthy diet rich in anti-inflammatory foods, regular physical activity, and adequate sleep can help mitigate neuroinflammatory processes and improve mood regulation.

### 5.4. Psychotherapy and Its Impact on Inflammation

Psychotherapy has emerged as a vital component in the treatment of depression, particularly in the context of neuroinflammation. As a non-pharmacological intervention, psychotherapy can influence inflammatory processes and contribute to improvements in mood and cognitive function. This chapter will explore the mechanisms by which psychotherapy impacts inflammation, the evidence supporting its efficacy in reducing inflammatory markers, and the implications for treating depression.

#### 5.4.1. Mechanisms of Psychotherapy on Inflammation

Psychotherapy, particularly cognitive behavioral therapy (CBT), has been shown to exert anti-inflammatory effects that may contribute to its therapeutic benefits in depression. The mechanisms underlying these effects may involve changes in stress perception, emotional regulation, and behavioral patterns. CBT can help individuals develop coping strategies to manage stress and negative emotions, which may reduce the activation of the HPA axis and the subsequent release of pro-inflammatory cytokines. Research by Dahl et al. found that patients who underwent psychotherapy exhibited normalized cytokine levels after treatment, suggesting that psychological interventions can lead to immunological changes that support recovery from depressive episodes [45]. Moreover, psychotherapy may enhance the resilience of individuals to stressors, thereby reducing the likelihood of inflammatory responses. By fostering adaptive coping mechanisms and promoting emotional regulation, psychotherapy can mitigate the impact of stress on the immune system [191]. This is particularly relevant in the context of chronic stress, which is known to activate inflammatory pathways and contribute to the development of depression [192]. It has been stated that individuals who engage in psychotherapy report lower levels of perceived stress, associated with reduced inflammatory markers [193].

#### 5.4.2. Clinical Evidence Linking Psychotherapy to Inflammatory Markers

A growing body of clinical evidence supports the association between psychotherapy and reductions in inflammatory markers in patients with depression. Moreira et al. conducted a randomized controlled trial that demonstrated a significant decrease in pro-inflammatory cytokines, including IL-6, following cognitive behavioral therapy in patients with major depressive disorder [192]. Research by Gunning-Dixon et al. indicated that elderly patients with depression who received psychotherapy showed improvements in both mood and inflammatory markers, including CRP [193]. The normalization of inflammatory markers following psychotherapy may also contribute to improved treatment outcomes and reduce the risk of relapse [194]. Furthermore, the impact of psychotherapy on neurotrophic factors, such as BDNF, has been observed in clinical studies. Research has shown that psychotherapy can increase BDNF levels, which may be associated with reductions in inflammation and improvements in mood [195,196]. This suggests that psychotherapy may not only address the symptoms of depression but also promote neurobiological changes that enhance resilience to stress and inflammation.

#### 5.4.3. Implications for Treatment

Given the role of inflammation in the pathophysiology of mood disorders, incorporating psychotherapy into treatment plans may enhance the overall efficacy of interventions. Combining psychotherapy with pharmacological treatments that target inflammation may provide a synergistic effect, improving outcomes for individuals with treatment-resistant depression [197,198]. Understanding the relationship between psychotherapy and inflammation may help identify individuals who are more likely to benefit from psychological interventions. Patients with elevated inflammatory markers may experience greater improvements in mood and inflammation following psychotherapy, suggesting that inflammatory profiles could serve as biomarkers for treatment response [199,200].

Overall, psychotherapy plays a significant role in modulating inflammation and improving mood in individuals with depression. By addressing both psychological and inflammatory aspects of depression, psychotherapy can contribute to improved mental health outcomes and enhance the overall well-being of individuals affected by mood disorders.

## 6. Conclusions

A complex network of immune responses, neurotransmitter alterations, and behavioral outcomes characterizes the interplay between neuroinflammation and depression. Understanding this relationship is crucial for developing more effective therapeutic strategies that address both the inflammatory and psychological components of mood disorders. Neuroinflammation can disrupt neurotransmitter systems critical for mood regulation. Furthermore, the activation of the HPA axis in response to stress can lead to increased glucocorticoid levels, further promoting neuroinflammatory processes and contributing to depressive symptoms. Factors such as genetic predisposition, history of trauma, and comorbid conditions (e.g., cardiovascular disease and obesity) may also influence an individual’s vulnerability to inflammation-induced depression. Psychological stress plays a significant role in the development and exacerbation of depression. The relationship between psychological stress and depression is characterized by a complex interplay of factors, including emotional regulation, stress perception, and resilience. The evidence suggests that not only does stress contribute to the onset and recurrence of depression, but depression can lead to increased stress generation, creating a vicious cycle that can be challenging to break. Interventions that focus on improving emotional regulation and resilience may offer promising avenues for reducing the impact of stress on depression.

Neuroinflammation is a multifaceted mechanism that significantly contributes to the pathophysiology of depression, involving complex interactions between cytokines, neurotransmitter systems, and neuroendocrine pathways. The interplay between pro-inflammatory cytokines, glial cell activation, and immune system dysregulation creates a conducive environment for the onset and persistence of depressive symptoms. Understanding these mechanisms may provide new avenues for therapeutic interventions aimed at modulating neuroinflammation to alleviate depression. Research has also shown that stress can exacerbate cytokine production, leading to persistent depressive-like behaviors. Therefore, the interplay between inflammatory processes and mood regulation highlights the potential for targeting cytokine pathways in the development of novel therapeutic strategies for depression, particularly in patients with comorbid inflammatory conditions.

Overall, the therapeutic implications of neuroinflammation in depression are profound, suggesting that anti-inflammatory strategies could complement existing treatments and improve outcomes for patients suffering from mood disorders. Future research should focus on identifying specific inflammatory markers and pathways that can be targeted therapeutically, as well as exploring the potential for personalized medicine approaches that consider individual inflammatory profiles.

## Figures and Tables

**Figure 1 ijms-26-01645-f001:**
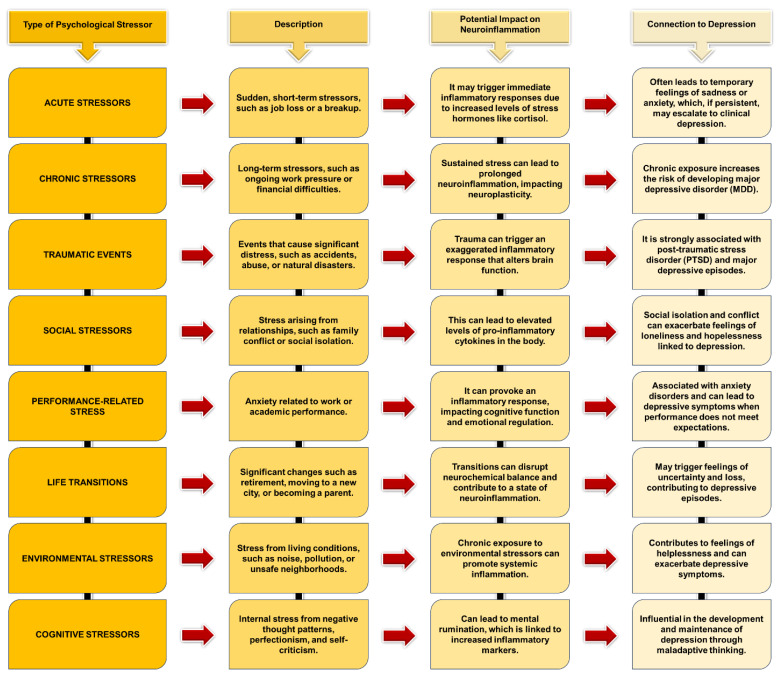
The relationship between types of psychological stressors and their impact on neuroinflammation and depression. One can observe how each type of stressor can uniquely influence the development and progression of depression, so understanding their interactions is essential for the development of effective interventions to prevent and treat depression.

**Figure 2 ijms-26-01645-f002:**
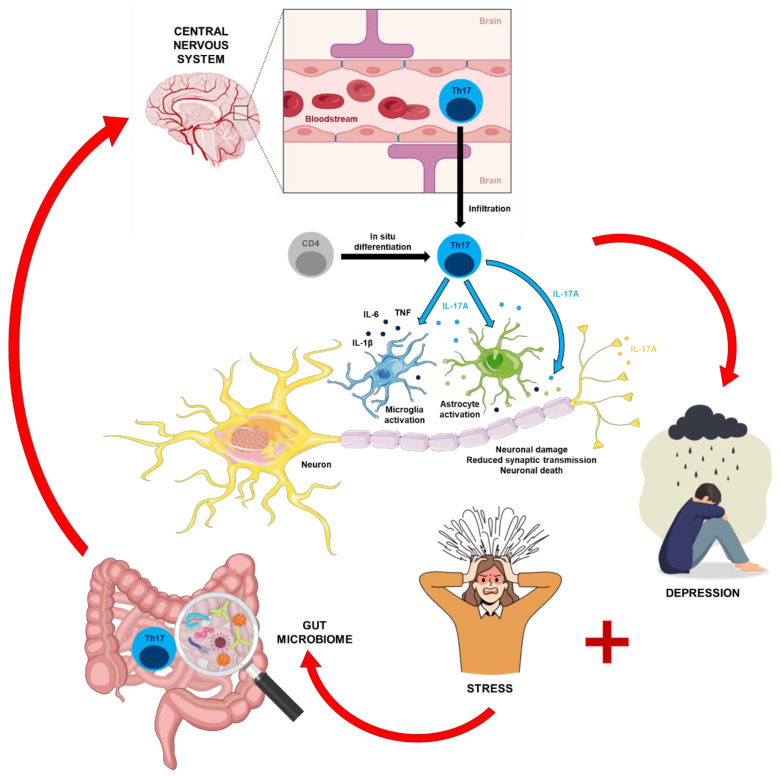
The gut–brain–immune axis and its role in neuroinflammation and mood disorders. T helper 17 cells (Th17) are a subset of pro-inflammatory T helper cells defined by the production of interleukin 17 (IL-17), also known as IL-17A; CD4—(cluster of differentiation 4), is a glycoprotein that serves as a co-receptor for the T-cell receptor; TNF—tumor necrosis factor; IL-6 and IL-1β—proinflammatory cytokines.

**Figure 3 ijms-26-01645-f003:**
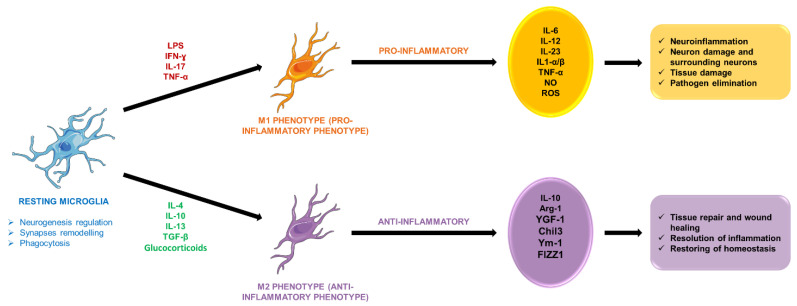
Microglial activation and polarization: a dichotomous response to stimuli in neuroinflammation and homeostasis. LPS—lipopolysaccharide; IFN-γ—interferon-gamma; TNF-α—tumor necrosis factor; TGF-β—transforming growth factor β; IL-6; IL-12; IL-23—cytokines pro-inflammatory; IL-10—cytokine anti-inflammatory; IL1-α/β—interleukin 1 α/β; NO—nitric oxide; ROS—reactive oxygen species; Arg-1—human gene; YGF-1—insulin-like growth factor 1; Chil3—chitinase-like protein 3; Ym-1—an eosinophilic chemotactic factor; FIZZ1—resistin-like molecule alpha.

**Figure 4 ijms-26-01645-f004:**
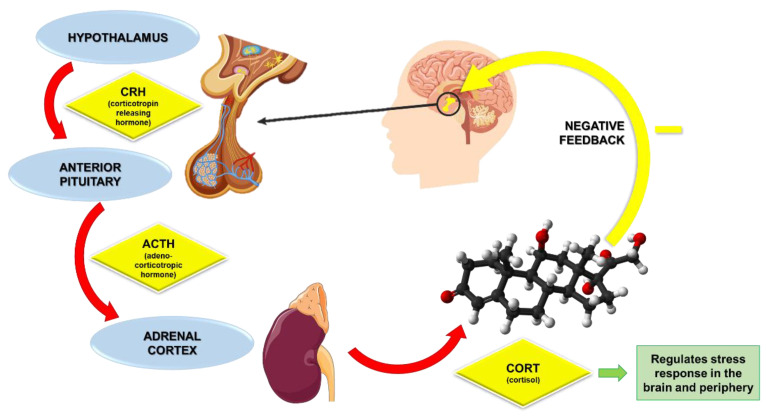
The hypothalamic–pituitary–adrenal (HPA) axis activation process. CRH—corticotropin-releasing hormone; ACTH—adrenocorticotropic hormone; CORT—cortisol.

## Data Availability

All the data used are included in the present review article. Further acquirements should be addressed to the corresponding author.

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
