# Peer review of "Unraveling the Complex Interplay Between Neuroinflammation and Depression: A Comprehensive Review"

_ijms, 2025, doi:10.3390/ijms26041645_

Round 1

Reviewer 1 Report

Comments and Suggestions for Authors

Sălcudean and colleagues provide a comprehensive review in which they discuss the complex interplay between neuroinflammation and depression. The Author concludes that understanding  this interplay is crucial for developing targeted therapies that can mitigate the effects of neuroinflammation on mood disorders. Despite the manuscript is well-written and interesting, some points should be addressed.

-       The manuscript is long. In order to make the manuscript more readable, the Authors should shorten it by deleting superfluous information and general concepts.

-       The Authors should also briefly discuss the role of biological sex in this context. In this respect, mood disorders are more common in women than in men (PMID: 30061743).

-       The Authors must avoid statements without references.

-       4.3. Lifestyle Factors: The beneficial effects of beverages enriched with anti-inflammatory  molecules on depressive symptoms should be also discussed. The Authors might want to report and discuss the following papers (PMID: 36829831; PMID: 34819888 and others).

-       The Authors must check for typos throughout the manuscript.

Author Response

Dear Reviewer 1, please find attached the response to your observations. Thank you.

Reviewer 2 Report

Comments and Suggestions for Authors

The authors of the manuscript (ijms-3461262) describe a bidirectional relationship between neuroinflammation and depression. The manuscript is generally well written, but some questions arise when reading:

1. Abbreviations used in figures must be explained in the legend below them, even if they are commonly known.

2. Line 191 "The glucocorticoid cascade hypothesis posits that chronic overproduction of glucocorticoids can damage glucocorticoid receptors (GRs), particularly in the hippocampus, leading to reduced negative feedback on the HPA axis and perpetuating a cycle of stress and depression [23]. " How GRs can be damaged, shortly describe the mechanism...e.g. caspase-related?

3. Line 257 "Pathway enrichment analyses have revealed that a significant proportion of genes associated with MDD are related to inflammatory responses, suggesting that immune dysregulation may be a fundamental aspect of the disorder." Give some examples of these genes

4. Line 368 "Variants in genes 368 associated with immune responses, such as TREM2 and CD33, have been linked to an increased risk of neurodegenerative diseases and depression" -Why? please, describe shortly the role of genes.

5. Line 557 "For example, exposure to chronic stress can lead to DNA methylation changes in genes associated with inflammation, which may contribute to the development of depressive symptoms" -which genes?

6. Line 623 "For example, variations in genes associated with the immune response, such as those encoding for cytokines or their receptors, can influence how individuals respond to environmental stressors and their subsequent risk for mood disorders [85]." Some examples of cytokine/receptor's genes need to be described.

7. Diet and lifestyle factors that are described twice (5.2 and 5.3 sections) need to be combined into one section e.g. section 5.2 can describe pharmacological interventions and 5.3 can cover modifiable lifestyle factors.

Author Response

Dear Reviewer 2, please find attached the response to your observations. Thank you.

Round 2

Reviewer 1 Report

Comments and Suggestions for Authors

The Authors have successfully addressed all the points I raised.